# Upper-plate conduits linked to plate boundary that hosts slow earthquakes

Ryuta Arai [1] ✉, Seiichi Miura [1], Yasuyuki Nakamura [1], Gou Fujie[1], Shuichi Kodaira [1], Yuka Kaiho[1], Kimihiro Mochizuki [2], Rie Nakata[2,3], Masataka Kinoshita[2], Yoshitaka Hashimoto [4], Yohei Hamada[5] & Kyoko Okino [6]

In shallow subduction zones, fluid behavior impacts various geodynamic processes capable of regulating slip behaviors and forming mud volcanoes. However, evidence of structures that control the fluid transfer within an overriding plate is limited and the physical properties at the source faults of slow earthquakes are poorly understood. Here we present high-resolution seismic velocity models and reflection images of the Hyuga-nada area, Japan, where the Kyushu-Palau ridge subducts. We image distinct kilometer-wide columns in the upper plate with reduced velocities that extend vertically from the seafloor down to 10–13 km depth. We interpret the low-velocity columns as damaged zones caused by seamount subduction and suggest that they serve as conduits, facilitating vertical fluid migration from the plate boundary. The lateral variation in upper-plate velocity and seismic reflectivity along the plate boundary correlates with the distribution of slow earthquakes, indicating that the upper-plate drainage system controls the complex pattern of seismic slip at subduction faults.

Plate boundaries in subduction zones portray a wide spectrum of slip behavior, from day-to-year-long slow slip events to instantaneous faulting, governed by diverse physical factors. Pore fluid is one of the most influential parameters of seismic slips, and pore fluid behavior has been intensively studied in recent years[1–3]. High fluid pressure is a prerequisite condition for slow earthquakes to nucleate at subduction faults[4]. At shallow parts of subduction zones (depth < 15 km), the pore fluid pressure is primarily controlled by the compaction/dehydration reactions of sediments entering the subduction zone[3]. Once released from the source, the fluid migrates upward via permeable zones into the forearc wedge owing to its buoyancy[5,6], forming mud volcanoes and cold seeps on the forearc basins and trench slopes[7]. Subducting seamounts (topographic highs at the plate boundary) impact the hydrological system in subduction zones by causing a heterogeneous distribution of permeability in the upper plate and forming a subduction channel filled with fluid-rich sediments[8–10]. However, the evidence on fluid distribution directly linked to subducting seamounts is limited. In particular, the along-strike extent of the impact of the subducting seamount is poorly constrained. In addition, our understanding of how the complex structure formed by subducting seamounts modulates the slip pattern at subduction faults remains unclear[11].

The Hyuga-nada subduction zone in southwest Japan is an excellent area to study the relationship between subduction earthquakes and forearc hydrology. Adjacent to the Nankai Trough, where megathrust earthquakes of magnitudes greater than 8 occur frequently, the Hyuga-nada area historically hosted several interplate earthquakes of magnitudes ~7[12,13], the source areas of which are delineated by long-term slow slip events on the downdip side[14] (Fig. 1a). On its updip side, slow earthquakes have been well-documented in recent

[1]Research Institute for Marine Geodynamics, Japan Agency for Marine-Earth Science and Technology, 3173-25 Showa-machi, Kanazawa-ku, Yokohama, Kanagawa 236-0001, Japan. [2]Earthquake Research Institute, The University of Tokyo, 1-1-1 Yayoi Bunkyo-ku, Tokyo 113-0032, Japan. [3]Lawrence Berkeley National Laboratory, Berkeley, USA. [4]Faculty of Science and Technology, Kochi University, Akebonocho 2-5-1, Kochi 780-8520, Japan. [5]Kochi Institute for Core Sample Research, Japan Agency for Marine-Earth Science and Technology, 200 Monobe Otsu Nankoku, Kochi 783-8502, Japan. [6]Atmosphere and Ocean Research Institute, The University of Tokyo, 5-1-5 Kashiwanoha Kashiwa, Chiba 277-8564, Japan. ✉e-mail: ryuta@jamstec.go.jp

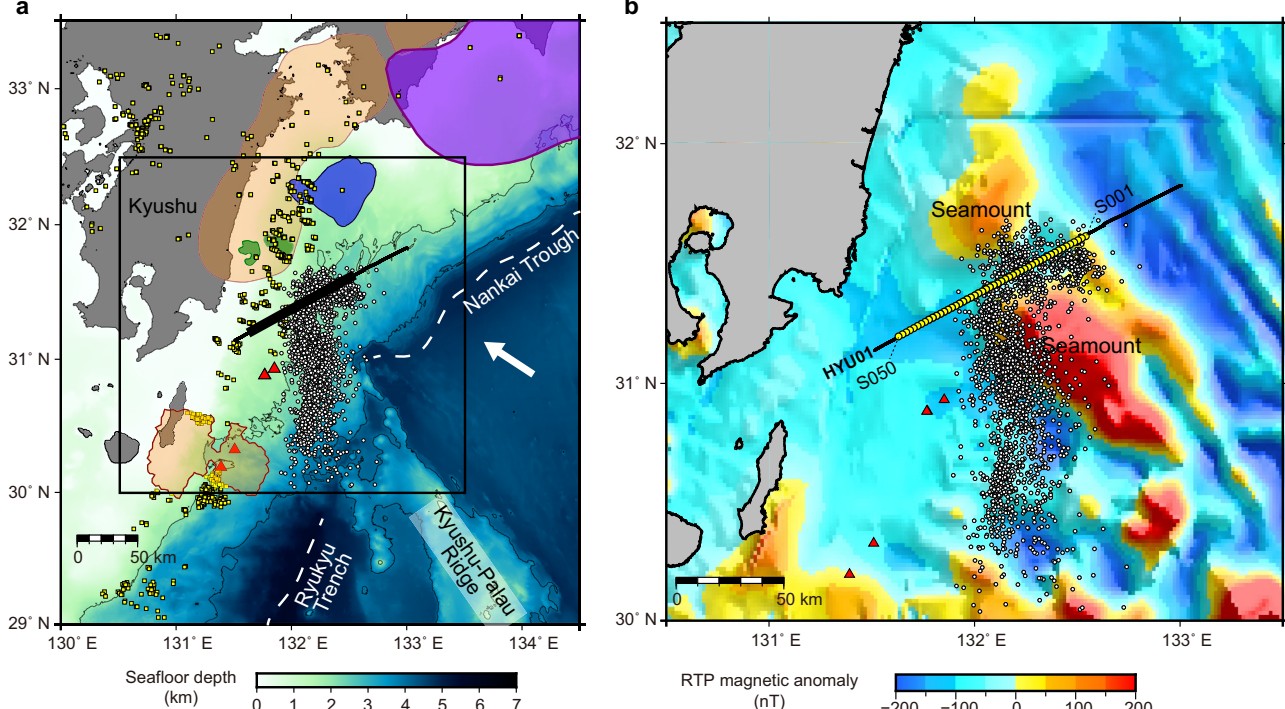

**Fig. 1 | Tectonic setting of the Hyuga-nada area. a** Regional bathymetric map of the study area. Purple, blue, and green shaded areas indicate the source areas of large megathrust earthquakes in 1946[47], 1968[12], and 1996[13], respectively Orange shades indicate the source areas of slow slip events[14,48]. White dots, yellow squares, and red triangles denote the locations of low-frequency tremors[14], repeating earthquakes[38], and mud volcanoes[20-22], respectively. The convergence direction of the Philippine Sea Plate relative to the Amurian plate is shown by the white arrow[28]. The thin black line indicates the seismic profile of this study, and the thick black line

indicates the range shown in Figs. 2, 4 and 5. The black box indicates the location enlarged in panel **b. b** Layout of the ocean bottom seismographs (OBSs) and multichannel seismic (MCS) reflection profiles superimposed on the reduced-to-pole (RTP) magnetic anomaly map[49]. The yellow circles indicate the locations of OBSs. The MCS reflection data were acquired along the black line. The velocity model and reflection images shown in Fig. 2 correspond to the 100-km-profile where the OBSs were deployed.

seafloor observations[15-18]. For example, the activity of low-frequency tremors exhibiting rapid epicentral migration (with a speed of tens to hundreds of kilometers per day)[15,16] implies that these events occur at the plate interface, driven by the spatiotemporal variation in the fluid pressure therein[19]. Another fluid-related phenomenon in the Hyuga-nada area is abundant mud volcanoes on the seafloor[20-22]. Although geochemical analyses of core samples suggested that these mud volcanoes are fed by the dehydration of clay minerals at shallow crustal depths[21], the origin of the fluid is poorly understood. Moreover, the hydrological system responsible for the transfer of deep fluids is unknown.

To better understand the geodynamic system in the Hyuga-nada subduction zone, we performed a dense seismic refraction experiment using ocean bottom seismographs (OBSs) and a multichannel seismic (MCS) reflection survey in the summer of 2020. For the refraction study, we laid out 50 OBSs at 2-km intervals on the seismic line (HYU01) (Fig. 1b). The seismic line, aligned parallel to the regional trend of the Nankai Trough in the SW-NE direction, crossed the source area of the low-frequency tremor distributed between the shallow and deep seamounts (Fig. 1). In this study, we construct a P-wave velocity (Vp) model by applying the advanced full waveform inversion (FWI) technique to the OBS data (Methods section). Our results provide important structural constraints of the crust from the seafloor down to the subducting plate at a significantly higher spatial resolution than previous studies[23-25]. We compare the Vp model with the reflection image on the coincident line for structural interpretations. Based on the seismic results, we suggest the origin of deep fluids and their pathways up to the seafloor mud volcanoes. We also compare them with the spatial distribution of slow and regular earthquakes to discuss the structural controls on their

occurrence. The structural characterization on the subduction megathrust faults using our study method can provide important information to evaluate the spatial extent of future megathrust earthquakes in the region.

## Results and Discussion
### Upper-plate heterogeneity and plate boundary structure
The FWI Vp modeling along the HYU01 profile reveals a remarkably heterogeneous structure in the upper plate (Fig. 2a). The most important finding is the lateral variation in the Vp, highlighted by horizontally alternating high- and low-velocity anomalies (Fig. 2b), which is supported by the varying apparent velocities of first-arrival refraction phases in the OBS record sections (Supplementary Figs. 1–8). We detected several low-velocity columns having Vp values < 3.5 km/s that extended near-vertically from the seafloor to depths of over 10 km. These upper-plate velocities are significantly lower than those in other areas of the Nankai subduction zone located at similar distances from the trench axis (~50 km)[25,26], suggesting that the subduction of the Kyushu-Palau Ridge increases the porosity of the forearc wedge over a broad region (about four times broader than the ridge itself). Interestingly, some of the low-velocity columns dip slightly toward the summit of the ridge located in the middle of the study area. Based on their fan-shaped geometry radiating from the seamount, we interpret that the low-velocity features represent an intensive fracture network; previous studies utilizing sandbox experiments explain that such networks consisting of subvertical faults can develop above subducting seamounts[27]. The dense distribution of fracture zones in the vicinity of the Kyushu-Palau Ridge may reflect the stress concentration in the region caused by seamount subduction, and their lower seismic velocities in the western part (compared with

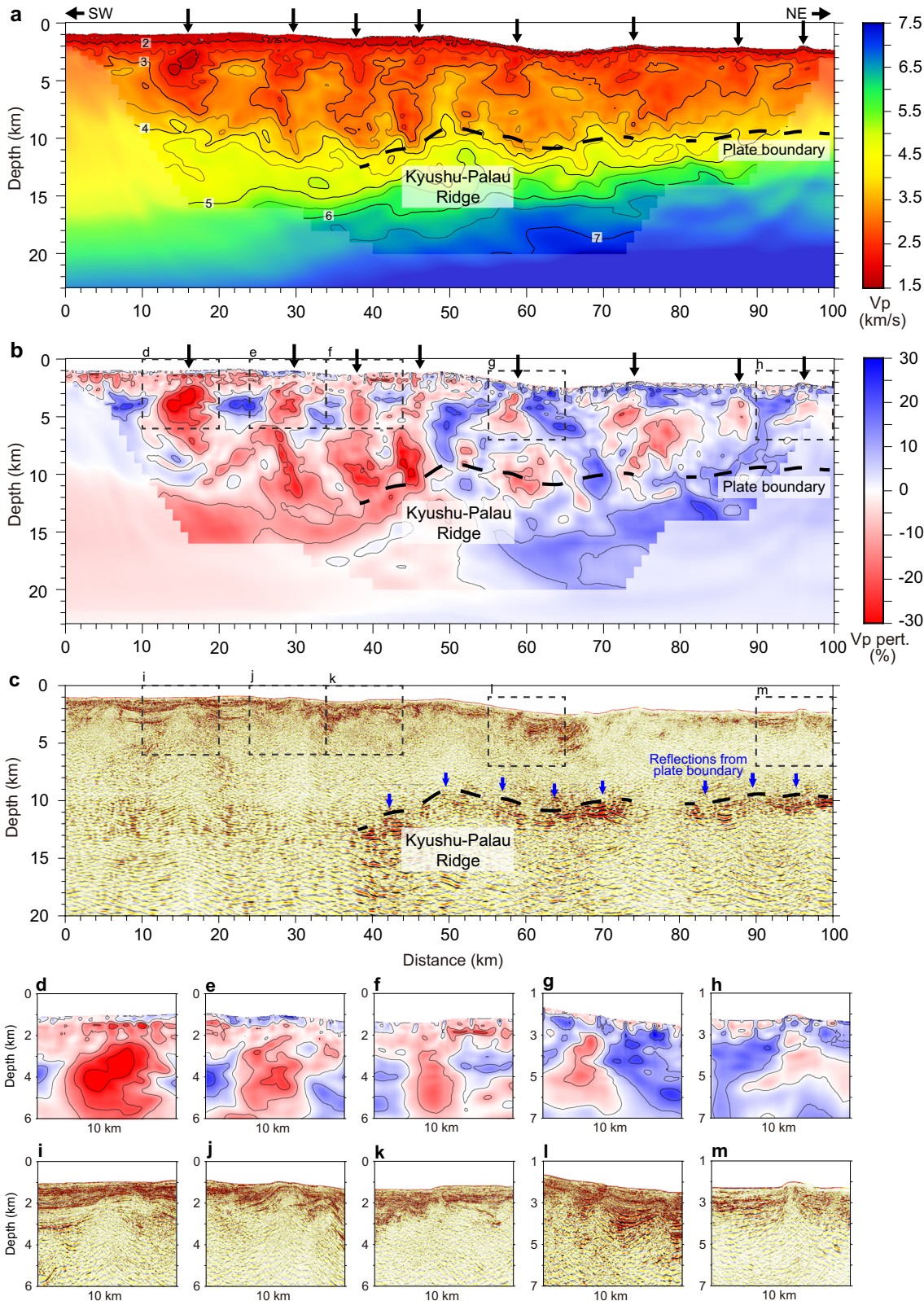

**Fig. 2 | Seismic structures along the HYU01 profile. a** Full waveform inversion (FWI) P-wave velocity (Vp) model highlighting the upper-plate lower-velocity columns (shown by black arrows). Areas with poor checkerboard recovery (Supplementary Fig. 10) are shaded. **b** Vp perturbation from the 1D depth average (depth from the seafloor). **c** Prestack depth-migrated seismic reflection image. **d**–**h** Close-up views of the low-velocity columns. **i**–**m** Close-up reflection images, with a focus on the domed structures atop the low-velocity columns.

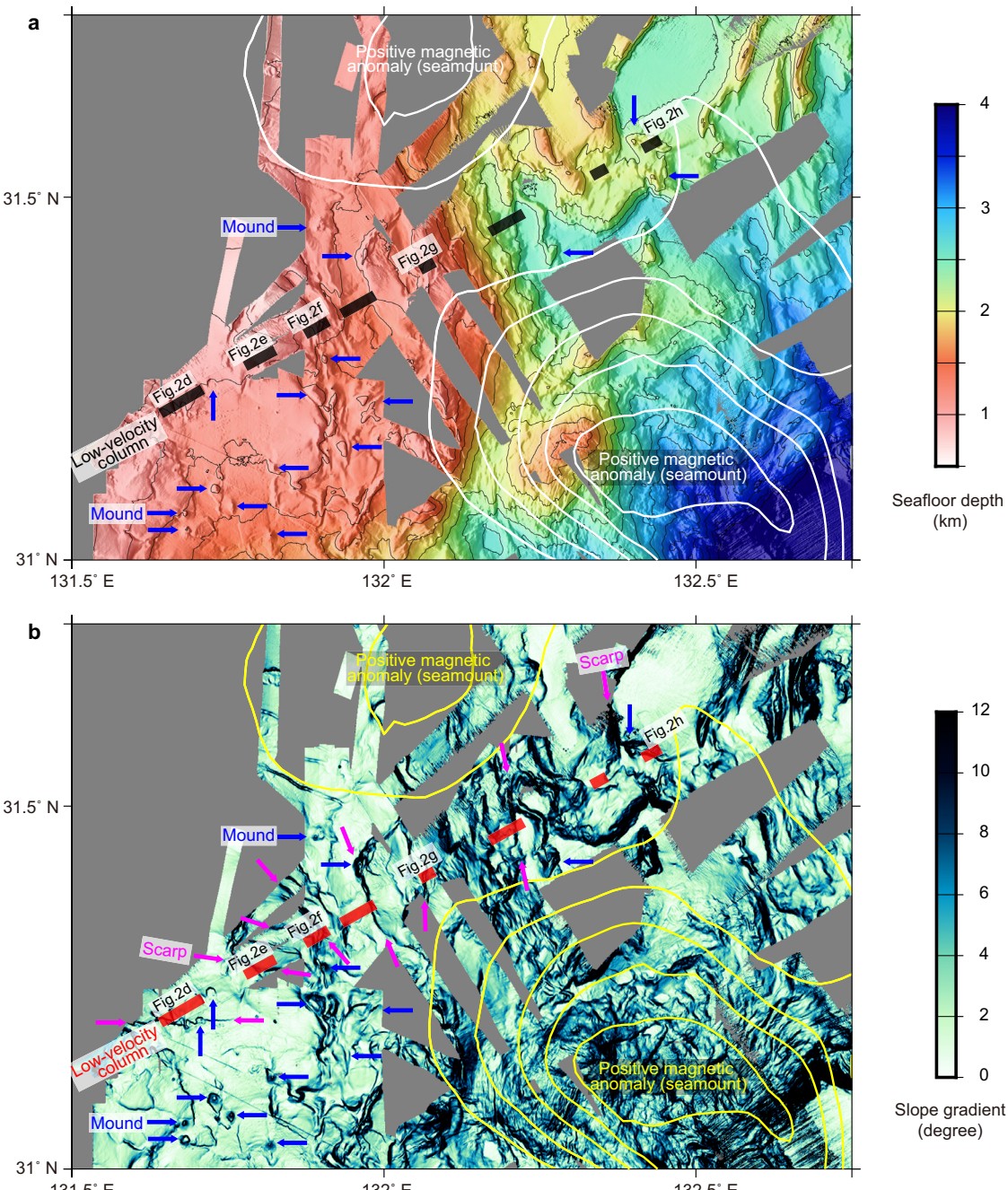

**Fig. 3 | Correlation of the low-velocity columns with seafloor morphology.**
Background colors represent seafloor depth (panel **a**) and slope gradient (panel **b**) highlighting circular mounds that possibly indicate mud volcanoes (blue arrows) and linear scarps (magenta arrows) that radiate from the seamounts (marked by white/yellow contour lines of the RTP magnetic data). The black/red thick lines indicate the location of the low-velocity columns in the upper plate (Fig. 2). To create this plot, we compiled the high-resolution bathymetry data obtained by multiple cruises using the Japan Agency for Marine-Earth Science and Technology (JAMSTEC) research vessels. The slope gradient of the seafloor was calculated using a generic mapping tool[50].

those in the eastern part; Fig. 2d–h) may indicate higher porosity and be attributed to the regional stress regime governed by the north-westward convergence of the incoming Philippine Sea Plate[28].

In enlarged views of the seismic reflection image, the upper-plate fracture zones form conical zones with atypical reflection patterns characterized by inverted-V-shaped reflectors (Fig. 2c). The horizontal reflectors corresponding to the sedimentary layers are highly disturbed and portray upwardly domed geometry (Fig. 2i−m) on the top of the low-velocity fracture zones (Fig. 2d–h). These seismic reflection features are typical of mud volcanoes[29] and suggest the occurrence of active upwelling flows inside the low-velocity columns. The seafloor

bathymetry of the area shows several circular mounds (with steep slopes) close to the seismic profile, indicating another typical feature of mud volcanoes (Fig. 3). Additionally, the data suggest an extremely complex pattern of fault development, characterized by several linear fault scarps. Based on the consistency with experimental studies[27], we suggest that these bathymetric features are associated with the ridge subduction. Importantly, the upper-plate low-velocity columns are situated where these linear scarps are developed (Fig. 3). These observational facts demonstrate that the upper plate accommodates well-developed fluid conduits that efficiently facilitate fluid flow to the seafloor and feed the mud volcanoes.

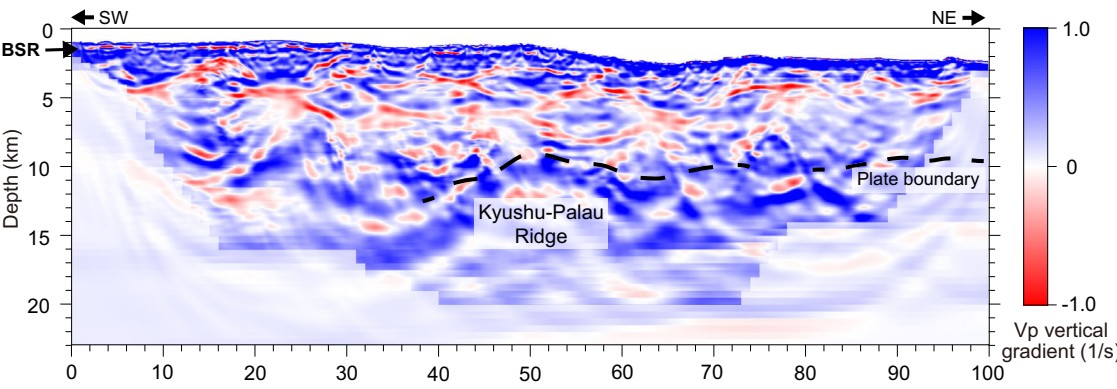

**Fig. 4 | Vertical gradient of the full-waveform inversion (FWI) Vp model.** This figure highlights velocity reversals, or low-velocity zones (marked by red-colored areas), at different depths. BSR indicates bottom simulating reflectors.

The seismic reflection image depicts reflective zones that extend laterally at depths of 10–13 km in the middle and eastern part of the seismic profile (40–100 km distance in Fig. 2c), with higher reflectivity in the flanks of the subducting ridge (40–45 km distance and 55–72 km distance in Fig. 2c) and away from the ridge (90–100 km distance in Fig. 2c). The reflective zones mostly trace the iso-Vp contour of 3.5–4.5 km/s (Fig. 2a) and also correspond to the areas having a positive velocity gradient in the depth direction, underlying those having a negative velocity gradient (Fig. 4). These layer structures probably represent low-velocity underthrust sediments atop the upper crust of the subducted Kyushu-Palau Ridge. In general, spatial variations in seismic reflectivity at the plate boundary is closely correlated with the physical properties and plate coupling conditions, which are highly influenced by in-situ fluid pressure[30,31]. At shallow depths where the temperature is below 150 °C, the pore fluid pressure is primarily controlled by the compaction and dehydration reactions of clay-rich sediments[3]. Notably, Vp values of 3.5–4.5 km/s imply that the reflective zones may be comprised of volcaniclastic sediments, which can hold a substantial amount of fluid before subduction and expel it when subducted[10,32,33]. Faint seismic reflections are observed in the western portion of the plate boundary. The disappearance of the reflection may suggest that fluids are starved there. Correspondingly, the westernmost low-velocity column is limited to the upper 8 km and unlikely to reach the plate boundary. The portion between the upper-plate low-velocity zones and the plate boundary increases the Vp to over 4.5 km/s and exhibits a sporadic reflection pattern with slightly higher reflectivity (0–40 km distance and 10–13 km depth in Fig. 2a and c).

### Fluid origin and migration system for mud volcano formation

Submarine mud volcanoes are a well-known feature in forearc basins and provide a window to explore the subsurface hydrological system in subduction zones[34]. One of the key issues regarding mud volcanoes is the diversity of the fluid origins. Even though previous studies state that the fluids originate from the dehydration of clay-rich sediments within the accretionary prisms, which typically occurs at depths of several kilometer by rapid sedimentation and compaction[34], recent studies report other potential sources, such as subducted sediments along deep thrust faults[35] and forearc mantle wedges[36]. The seismic data recorded in our study offer new evidence for the previously derived conclusions that the fluid source for the mud volcanoes in the Hyuga-nada subduction zone is along the plate boundary at depths >10 km.

At the shallow parts of subduction zones, fluid is likely to migrate via the decollement and branching faults, because they act as the most permeable pathways for fluid flow. A significant difference in the fluid feeding system in the Hyuga-nada area from this conventional concept is that the fluids may be ascending upward through the near-vertical conduits. We suggest that the fracture network in the Hyuga-nada subduction zone is primarily paved by the subducting seamount; the Vp model indicates that the network consists of discrete low-velocity columns that are densely distributed atop and around the subducting ridge (Fig. 2a and b). Our results also reveal some velocity reversals at different depths within the upper plate (Fig. 4). These results suggest that fluids from the plate boundary may be captured and accumulate at multiple depths within the upper plate and/or the upper plate itself also generates some fluids by dehydration of clay-rich sediments[37]. The latter possibility is also consistent with the observation that the upper-plate conduits are linked to not only the fluid-rich plate boundary (40–72 km and 80–100 km distances in Fig. 5a) but also the fluid-starved portions (30–40 km and 72–80 km distance in Fig. 5a).

### Drainage system and its control on slow earthquakes

The variations in the Vp and reflectivity along the plate boundary agree well with the distribution of tremors and very low-frequency earthquakes (VLFEs) in the region. For example, while tremors and VLFEs actively occur around the subducting Kyushu-Palau Ridge, where the plate boundary is reflective and exhibits low seismic velocities (Vp of 3.5–4.5 km/s), their activities disappear in the western part of the profile (0–40 km distance in Fig. 5b), corresponding to the increase in the Vp value (> 4.5 km/s) and the waning seismic reflectivity around the plate boundary (0–40 km distance in Fig. 5a). Instead, the western part of the plate boundary is dominated by repeating regular earthquakes[38] (Figs. 1 and 5b). This indicates a rapid transition in the slip pattern along the plate boundary that corresponds well with the variations in the physical properties around the plate boundary faults. As the Vp value measured in this study at the transitional area (~4.5 km/s) is consistent with that in previous studies on other subduction zones[39,40], the value of 4.5 km/s is considered a common threshold to characterize the slip behavior along the megathrust faults.

Several episodes of tremors and VLFEs are observed within a localized area of an ~30-km range[15–17] ("Isolated" tremor/VLFE episodes at 40–70 km distance in Fig. 5) and, similar to the western limit of the tremor zone, the eastern limits of these episodes coincide well with the nonreflective portion of the plate boundary (72–80 km distance in Fig. 2c). On the other hand, this nonreflective portion sometimes hosts tremor activities ("Pervasive" tremor episodes in Fig. 5). Although factors controlling this variation in tremor occurrence are unclear, one possibility inferred from our structural observation is the role of the upper-plate conduit. The Vp model indicates that one of the low-velocity anomalies exits in the upper plate just above the non-reflective portion (at 72–80 km distance in Figs. 2b and 5a) and thus may suggest that the fluid pressure at the plate boundary is modulated by the partial leakage of fluids into the upper-plate conduits. In general, the

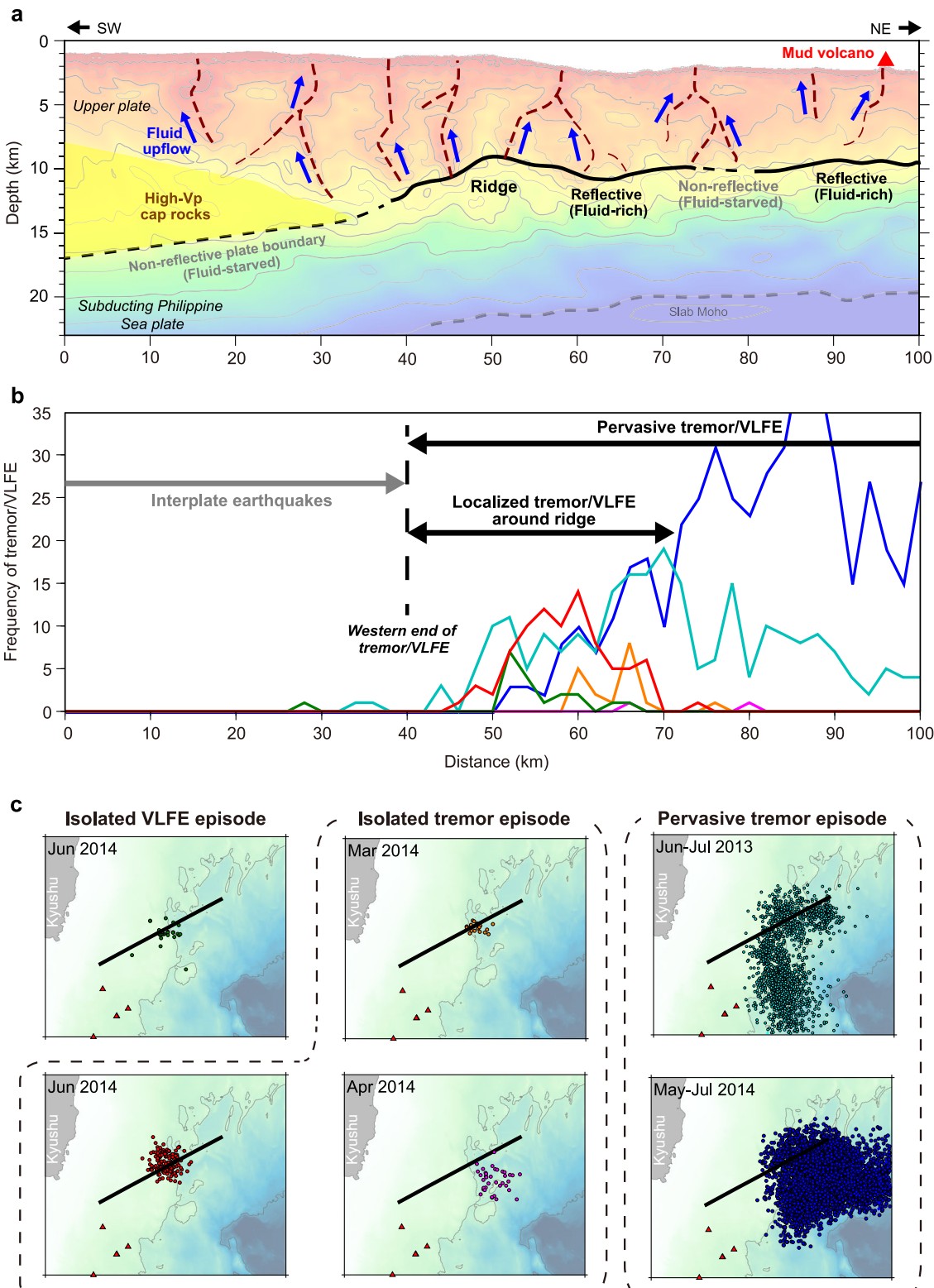

**Fig. 5 | Comparison of structural characteristics with the seismogenic patterns along the plate interface. a** Interpretations of the upper-plate fluid conduits and seismic reflectivity at the plate boundary. The background color is the Vp model in Fig. 2a. Iso-velocity contour of 7.5 km/s is used as a proxy for the slab Moho. **b** Frequency plot of tremors[15,16] and very low-frequency earthquakes (VLFE)[17]. The numbers of events within 5-km distance from the seismic profile are projected in the figure. Colors of each curve corresponds to different tremor/VLFE episodes shown in panel **c**. **c** Map-view distribution of each episode of tremors and VLFEs (colored dots). Solid lines indicate the location of the HYU01 seismic profile.

upper-plate drainage system is known to play such a role[4]. Although this hypothesis is proposed for the tremor activity in deep subduction zones (i.e., depths > 30 km), it may also be true for shallow subduction zones (i.e., depths < 15 km).

## Methods

### Seismic data acquisition

The seismic data were acquired through two research cruises conducted in August 2020. During the KM20-05 cruise by the R/V Kaimei of Japan Agency for Marine-Earth Science and Technology (JAMSTEC), we deployed 50 OBSs along the HYU01 profile at 2-km intervals. Each OBS contained three-component geophone and hydrophone sensors (with a natural frequency of 4.5 Hz; the sampling rate for recording was 200 Hz). The OBSs recorded the acoustic signals from 596 airgun shots triggered every 200 m along the 120-km-long seismic line. All the OBSs were recovered by the R/V Kairei during the KR20-10 cruise, providing the seismic records of the airgun signals. We also collected the MCS reflection data on the same line as the OBS refraction survey, using a 5.6-km-long 444-channel streamer cable onboard the Kaimei; the shooting interval for the reflection survey was 50 m. For both the surveys, the Kaimei's tuned airgun arrays (with a total air volume of 10,600 cubic inch) were used as the source.

### Vp model construction from OBS refraction data

After the recovery, the OBS data were preprocessed for further analyses. First, we located the OBS positions at the seafloor, using the same method applied in our previous study[41]. Then, we determined the first breaks in the refraction waves manually, using the vertical component of the geophone data. The picking error ranging from 30 ms to 60 ms were assigned to each pick, depending on the offset and the noise level of the data. A total of 27,902 first arrivals were used for a traveltime tomography analysis[42] to derive a preferable Vp model. The initial Vp model for the first-arrival traveltime tomography (FAT) was created by referring to the seismic refraction studies close to the survey line considered in this study[25] (Supplementary Fig. 9a). The final Vp model developed using FAT that explained the traveltime data within the error range (root mean square [RMS] traveltime residual of 56 ms) was obtained after 15 iterations of inversion (Supplementary Fig. 9b).

To improve the resolution of the Vp model, we further applied the FWI analysis to the OBS records. For the analysis, we used TOY2DAC, an acoustic 2D and frequency-domain FWI code developed by SEISCOPE[26,43]. For the modeling, the Vp structure of 100 × 23 km was gridded every 40 m. The result of FAT was used in the starting model. We applied the pre-processing to the vertical geophone data, such as muting before first arrival and time-damping with different time windows at each step of inversion. To stabilize the calculation, the inversion was initiated from a low frequency band of 2.5–3.5 Hz at the first step and then sequentially increased up to a higher frequency band of 2.5–7.5 Hz at the final step. We sampled the frequencies every 0.25 Hz within each frequency band. Other inversion parameters are summarized in Supplementary Table 1. The OBS records contained significant attenuations in the refraction phases, with shadow zones at some locations. To better reproduce these features in the OBS records, we introduced Vp-dependent attenuation factors (Q values) in the calculation. We used the Q values of 10,000, 50, 90, 350, 500, and 850 for the regions having Vp values < 1.55, 1.55–3.5, 3.5–4.5, 4.5–6.5, 6.5–7.5, and > 7.5 km/s, respectively[44,45]. The initial density model was converted from the Vp model, using the Gardner relation[46]. Both the Q and density values remained unchanged during inversion, and only the Vp values were updated. The overall waveform fits (Supplementary Fig. 1–8), reduction in the cost function (Supplementary Table 1), and consistency with the MCS reflection data were carefully examined to evaluate the reliability of the final model. The checkerboard resolution tests demonstrated that the FWI analyses was capable for resolving the velocity anomalies with 1.5 km (horizontally) x 1 km (vertically) down to the depth of ~ 15 km (Supplementary Fig. 10).

### MCS reflection data processing

The primary processing of the MCS reflection data was carried out by DownUnder GeoSolutions (DUG). The data were processed following a standard prestack flow that included trace editing, tidal static correction, and low-cut filtering (of 1–3 Hz). The processing also included swell-noise attenuation, deghosting, source signature deconvolution, 2D surface-related multiple elimination, least-squares adaptive subtraction, and high-resolution radon demultiple for noise reduction. Velocity analyses and prestack Kirchhoff depth migration were performed to produce the final seismic reflection image (Fig. 2c).

### Magnetic data processing

The magnetic anomaly data were downloaded from the Marine Trackline Geophysical Database of the National Centers for Environmental Information (https://www.ngdc.noaa.gov/mgg/gdas/). The data were gridded at 1arc-min (bin size), and then were transformed to the vertical component of the magnetic field produced by the same source magnetized in the vertical direction [reduced-to-pole (RTP) anomaly]. The paleo latitude of the study area was assumed to be 26° N, while considering the northward convergence direction of the Philippine Sea Plate.

## Data availability

The OBS and MCS data that support the conclusions are available through the JAMSTEC Seismic Survey Database (http://www.jamstec.go.jp/obsmcs_db/e/).

## Code availability

FWI code used in this study is available from SEISCOPE Consortium (https://seiscope2.osug.fr). GMT code is available at the website of University of Hawaii (http://www.soest.hawaii.edu/gmt/).

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

## Acknowledgements

We thank the captains, ship crews, and onboard technicians of the R/V Kaimei and Kairei for their dedicated efforts in data acquisition during the KM20-05 and KR20-10 cruises. Discussions with A. Gorszczyk, T. Tonegawa, K. Shiraishi, M. Otsubo, and M. Furuichi were helpful in improving the construction of the Vp model and interpreting its results. Additionally, we are grateful to R. Miura and K. Takizawa for their assistance in MCS data processing. This study was supported by the Japan Society for the Promotion of Science (JSPS) KAKENHI program (Grant numbers: 16H06475 (S. M.), 19H04629, 21H05202, and 22K03789 (R.A.)).

## Author contributions

R.A. analyzed the OBS and MCS reflection data, interpreted the results, and drafted the paper through the discussion with all the co-authors. S.M. and Y.N. organized the research cruises of KM20-05 and KR20-10 and served as chief scientists. G.F., S.K., and K.M. supervised the data acquisition and processing. Y.K. participated in the data-acquisition cruise. R.N., M.K., Y. Has., and Y. Ham. led the acquisition and processing of the MCS reflection data. K.O. analyzed the seafloor magnetic data. All the authors contributed to discussions of the study results.

## Competing interests

The authors declare no competing interests.
