## [Peer Review File · Nature Communications]

REVIEWER COMMENTS

Reviewer #1 (Remarks to the Author):

This manuscript presents a high-resolution two-dimensional seismic velocity model along the forearc wedge of the Hyuga-nada subduction zone obtained from full waveform inversion (FWI) of wide-angle ocean bottom seismic data. The model illuminates sub-vertical low velocity zones (LVZs) that the authors interpret as damaged and fractured zones in the upper plate created by subduction of a seamount that form a fluid drainage network that controls the hydrogeology system in this region. By comparing the seismic model with seafloor bathymetry, magnetic data, and non-volcanic tremor and very-low-frequency earthquakes, the authors make a convincing case that the LVZs represent a fluid drainage zones that impact the plate interface seismogenic properties.

Understanding fluid migration and retention in subduction settings is topic of the greatest importance because of the major role that fluids play in subduction plate interface properties, with implication for seismic hazards. In this context, this manuscript will be an important contribution in this field and it would be an appropriate publication for Nature Communications.

The manuscript is succinct, well written, with nice and clear illustrations, and makes a convincing case for most of the interpretations. However there are some aspects that require improvement before acceptance, in particular more details about the FWI modeling approach, and discussion about some of the interpretations that are apparently not fully supported by the data.

Below is a list of all my comments, both major, minor, and editorial, in order of appearance in the manuscript. Numbers in my comments refer to numbered manuscript lines in the submitted PDF document.

Lines 45-46. Suggested rewording: "We image distinct several-kilometer-wide columns in the upper plate with reduced velocities that extend vertically from the seafloor up to 10-13 km depth."

47. Change "radiated" to "radiate", and "were" to "are".

51. Change "correlates to" to "correlate with".

66. Change “high” to “highs”.

75. Delete “huge”.

99. Change “the identical” to “a coincident”.

114. Change “near-vertically to the depths” to “near-vertically from the seafloor to depths”.

116. Change “that have the same distance from” to “located at similar distances from”.

117-118. Suggested rewording: “...Kyushu-Palau Ridge increases the porosity of the forearc wedge over a broad region (about four times broader than the ridge itself).”.

118. Change “incline” to “dip”.

123. Delete “the” before fracture, and change “just above” to “in the vicinity”.

125. Please define how “maturity” is used here. How does Vp or geometry of fractured zones inform about maturity?

132. Change “which indicate a great correlation with” to “which correlate spatially with”.

145. This sentence indicates that reflectivity is higher at ~40-60 km model distance. However reflectivity over the subducted seamount (~45-55 km) is actually lower than in other regions. Reflectivity is stronger away from the ridge at ~64-72 km and 90-100 km.

149. Accreted sediments?

150. Delete “the” preceding “spatial” and “seismic”. Change “variation” to “variations”.

151. Change “correlated to” to “correlated with”.

155, 156. Change “reflective zones are” to “reflective zones may be”, and “sediments that can” to “sediments, which can”.

159-162. These sentences seem to imply that low plate reflectivity in the western section indicates fluid starvation, and it is reflected by LVZs being limited to the upper section. However the LVZ near 30 km model distance is overlaid by a narrow vertical LVZ extending to 12 km depth where plate boundary reflectivity is low. This structure seems to contradict the interpretation given in these sentences.

172. Change “offer a new implication on” to “offer new evidence for”.

181. Change “consisted of independent” to “consists of discrete”.

206-210. The wording in these sentences is confusing. Please indicate which LV anomaly exists in the upper plate above the non-reflective interface (the anomaly shown in Fig. 2e, I presume?). Are the authors implying that efficient drainage of fluids through this anomaly reduces pore pressure and creates a barrier to tremor migration? If that is what is being suggested here, then it fails to explain why the anomaly at ~70-80 km, which is draining fluids from a non-reflective section of the plate interface (fluid starved, accordingly to Fig. 5a), does not act as a barrier to tremor in this section of the profile.

258. Change “were applied to the traveltimes” to “were used for a traveltimes”.

259. Reference 45 describes a traveltimes mapping method (i.e., a wide-angle pre-stack depth migration), not a traveltimes tomography method. Please give details or cite reference for the FAT analysis.

269. Please indicate which data were used for the FWI: vertical geophone or hydrophone? Please also indicate it in the captions of Extended Data Figures 1-8.

269-270. Please show in Extended Data Figures 1-8 the windows and mutes used so readers can see which portions of the record sections were used in the FWI.

271. Are 2.5 and 7.5 Hz the only two frequencies used? Authors should show a graph of the data spectra.

275-276. Based on the V_p - Q values listed here, the authors have used $Q=90$ for $V_p=3.5-4.5$ km/s. However in cited reference 47, $Q=349$ for $V_p=3.6-4.6$ km/s. Please explain the discrepancy, and comment on any potential implications for the FWI results.

279. Change “inverted.” to “updated.”. Authors should also show a figure illustrating the reduce in cost function.

293. Please include the pre-stack depth migration velocity model in Extended Data figure 9.

Figure 5a. In the caption, indicate that background color is V_p as in Fig. 1. The slab Moho dashed line shown in this figure has not been shown in previous figures nor mentioned in the text. Cite reference for the source of this Moho interface.

Reviewer #2 (Remarks to the Author):

This manuscript connects the occurrence of slow earthquakes with the presence of fluids in the upper plate. Some structures observed in the seismic image are interpreted as fluid migration paths based on the correlation with low-velocity zones observed in the velocity model and with the location of mud volcanoes. The zones interpreted as fluid-rich broadly correlate to the spatial distribution of tremors and very low-frequency earthquakes.

The manuscript is well-written and explanatory. However, the influence of the fluid distribution in the seismogenic behaviour of subduction zones is already known and it is described in different publications (see a review in Saffer and Tobin, 2011; and Philip et al., 2023 for an example of a recent study in the Cascadia subduction zone). The changes in the reflectivity of the plate boundary reflector on seismic images were described in Ranero et al. (2008), and this reference should be included in the manuscript.

Thus, the main novelty of these results may be the direct comparison with the recorded seismicity in the area. However, the comparison of the seismic images with the earthquake location is only done spatially. For example, in Figure 5c, earthquakes within a large area are included, but the seismic profile shows high variability in the structure of the fluid migration system in distances <15 km. This variability in the structure should be integrated into the discussion of the seismogenic behaviour. To strengthen

the results, I suggest focusing on the earthquakes occurring close to the seismic profile, and locating these earthquakes along the profile, to compare the crustal structure with the seismicity of the area. Also, nothing is mentioned about the depth of nucleation of these earthquakes. Are they nucleated at the plate boundary or at upper crust depths? If they are located at upper crust depths, why is there a difference driven by the fluid content of the plate boundary?

Regarding the fluid's interpretation, there are fluids interpreted in the upper crust of the entire seismic profile, although there are only interpreted at the plate boundary towards the NE. The correlation of the low-velocity zones observed in the velocity model with mud volcanoes is unclear to me (Figure 3, see comments below). In my opinion, further discussion of the fluid origin and its implication for the seismogenic behaviour is needed.

Regarding the figures, I suggest revising Figure 3 to make it easier to understand. As it is, the relation of the low-velocity column with the mud volcanoes is difficult to see. Are there any mud volcanoes at the exact location of the low-velocity columns?

In Figure 1 a, I can't find the green shaded area.

Suggested bibliography:

Philip, B. T., Solomon, E. A., Kelley, D. S., Tréhu, A. M., Whorley, T. L., Roland, E., Tominaga, M., & Collier, R. W. (2023). Fluid sources and overpressures within the central Cascadia Subduction Zone revealed by a warm, high-flux seafloor seep. *Science Advances*, 9(4). <https://doi.org/10.1126/sciadv.add6688>

Ranero, C. R., Grevemeyer, I., Sahling, H., Barckhausen, U., Hensen, C., Wallmann, K., Weinrebe, W., Vannucchi, P., Von Huene, R., & McIntosh, K. (2008). Hydrogeological system of erosional convergent margins and its influence on tectonics and interplate seismogenesis. *Geochemistry, Geophysics, Geosystems*, 9(3). <https://doi.org/10.1029/2007GC001679>

Saffer, D. M., & Tobin, H. J. (2011). Hydrogeology and mechanics of subduction zone forearcs: Fluid flow and pore pressure. *Annual Review of Earth and Planetary Sciences*, 39, 157–186. <https://doi.org/10.1146/annurev-earth-040610-133408>

Point by point response to reviewer comments. Our responses are in **red bold**.

Reviewer #1

This manuscript presents a high-resolution two-dimensional seismic velocity model along the forearc wedge of the Hyuga-nada subduction zone obtained from full waveform inversion (FWI) of wide-angle ocean bottom seismic data. The model illuminates sub-vertical low velocity zones (LVZs) that the authors interpret as damaged and fractured zones in the upper plate created by subduction of a seamount that form a fluid drainage network that controls the hydrogeology system in this region. By comparing the seismic model with seafloor bathymetry, magnetic data, and non-volcanic tremor and very-low-frequency earthquakes, the authors make a convincing case that the LVZs represent a fluid drainage zones that impact the plate interface seismogenic properties.

Understanding fluid migration and retention in subduction settings is topic of the greatest importance because of the major role that fluids play in subduction plate interface properties, with implication for seismic hazards. In this context, this manuscript will be an important contribution in this field and it would be an appropriate publication for Nature Communications.

The manuscript is succinct, well written, with nice and clear illustrations, and makes a convincing case for most of the interpretations. However there are some aspects that require improvement before acceptance, in particular more details about the FWI modeling approach, and discussion about some of the interpretations that are apparently not fully supported by the data.

Below is a list of all my comments, both major, minor, and editorial, in order of appearance in the manuscript. Numbers in my comments refer to numbered manuscript lines in the submitted PDF document.

Lines 45-46. Suggested rewording: “We image distinct several-kilometer-wide columns in the upper plate with reduced velocities that extend vertically from the seafloor up to 10-13 km depth.”.

We modified as suggested (Lines 45-47 in the revised manuscript with track changes).

47. Change “radiated” to “radiate”, and “were” to “are”.

We deleted this part in the process of reducing the volume of abstract.

51. Change “correlates to” to “correlate with”.

We replaced “to” with “with” (Line 53 in the revised manuscript with track changes).

66. Change “high” to “highs”.

We modified as suggested (Line 68 in the revised manuscript with track changes).

75. Delete “huge”.

We modified as suggested (Line 77 in the revised manuscript with track changes).

99. Change “the identical” to “a coincident”.

We replaced “identical” with “coincident” (Line 101 in the revised manuscript with track changes).

114. Change “near-vertically to the depths” to “near-vertically from the seafloor to depths”.

We modified as suggested (Line 116 in the revised manuscript with track changes).

116. Change “that have the same distance from” to “located at similar distances from”.

We modified as suggested (Line 118 in the revised manuscript with track changes).

117-118. Suggested rewording: “...Kyushu-Palau Ridge increases the porosity of the forearc wedge over a broad region (about four times broader than the ridge itself).”.

We modified as suggested (Lines 120-121 in the revised manuscript with track changes).

118. Change “incline” to “dip”.

We modified as suggested (Line 122 in the revised manuscript with track changes).

123. Delete “the” before fracture, and change “just above” to “in the vicinity”.

We modified as suggested (Line 127 in the revised manuscript with track changes).

125. Please define how “maturity” is used here. How does Vp or geometry of fractured zones inform about maturity?

We agree that the term “maturity” was ambiguous here. We intended to suggest that the western part has lower Vp values and thus higher porosity than eastern part. We modified in that way (Lines 129-131 in the revised manuscript with track changes).

132. Change “which indicate a great correlation with” to “which correlate spatially with”.

We modified as suggested (Lines 137-138 in the revised manuscript with track changes).

145. This sentence indicates that reflectivity is higher at ~40-60 km model distance. However reflectivity over the subducted seamount (~45-55 km) is actually lower than in other regions. Reflectivity is stronger away from the ridge at ~64-72 km and 90-100 km.

Thank you for this comment. We revised the explanation accordingly (Lines 150-152 in the revised manuscript with track changes).

149. Accreted sediments?

We added “underthrust” before “sediments (Line 155 in the revised manuscript with track changes).

150. Delete “the” preceding “spatial” and “seismic”. Change “variation” to “variations”.

We modified as suggested (Lines 156-157 in the revised manuscript with track changes).

151. Change “correlated to” to “correlated with”.

We modified as suggested (Line 157 in the revised manuscript with track changes).

155, 156. Change “reflective zones are” to “reflective zones may be”, and “sediments that can” to “sediments, which can”.

We modified as suggested (Line 162 in the revised manuscript with track changes).

159-162. These sentences seem to imply that low plate reflectivity in the western

section indicates fluid starvation, and it is reflected by LVZs being limited to the upper section. However the LVZ near 30 km model distance is overlaid by a narrow vertical LVZ extending to 12 km depth where plate boundary reflectivity is low. This structure seems to contradict the interpretation given in these sentences.

“the westernmost low-velocity column” (Line 167 in the revised manuscript with track changes) indicates not the LVZ near 30 km model distance but one at 15-20 km model distance. A body with V_p over 4.5 km/s obviously underlies the LVZ at 15-20 km model distance at depths greater than 8 km and thus the LVZ is unlikely to reach the plate boundary as described in the text (Lines 167-168 in the revised manuscript with track changes). It is unclear whether the LVZ near 30 km model distance reach the plate boundary since the same V_p body exists at the bottom of the LVZ (See our interpretations in Figure 5a).

172. Change “offer a new implication on” to “offer new evidence for”.

We modified as suggested (Lines 178-179 in the revised manuscript with track changes).

181. Change “consisted of independent” to “consists of discrete”.

We modified as suggested (Line 187 in the revised manuscript with track changes).

206-210. The wording in these sentences is confusing. Please indicate which LV anomaly exists in the upper plate above the non-reflective interface (the anomaly shown in Fig. 2e, I presume?). Are the authors implying that efficient drainage of fluids through this anomaly reduces pore pressure and creates a barrier to tremor migration? If that is what is being suggested here, then it fails to explain why the anomaly at ~70-80 km, which is draining fluids from a non-reflective section of the plate interface (fluid starved, accordingly to Fig. 5a), does not act as a barrier to tremor in this section of the profile.

To avoid this confusion, we revised the text to clearly indicate that the LV anomaly mentioned here is the one at 70-80 km distance (not the western one in Fig. 2e; Lines 215-216 in the revised manuscript with track changes). As mentioned in the text (Lines 209-213 in the revised manuscript with track changes) and shown in Fig. 5, 70-80 km distance in the profile corresponds to the eastern end of some of the tremor episodes and thus seems to act as a barrier.

258. Change “were applied to the travelttime” to “were used for a travelttime”.

We modified as suggested (Line 267 in the revised manuscript with track changes).

259. Reference 45 describes a travelttime mapping method (i.e., a wide-angle pre-stack depth migration), not a travelttime tomography method. Please give details or cite reference for the FAT analysis.

We replaced reference 45 with Fujie et al. (2013) (new reference 46; Lines 504-509 in the revised manuscript with track changes).

269. Please indicate which data were used for the FWI: vertical geophone or hydrophone? Please also indicate it in the captions of Extended Data Figures 1-8.

We used vertical geophone data. We added the description to Line 279 and captions of Supplementary Figures 1-8.

269-270. Please show in Extended Data Figures 1-8 the windows and mutes used so readers can see which portions of the record sections were used in the FWI.

We added an additional figure showing the offset range and the time windows used in the FWI to the Supplementary Information File (Supplementary Figure 11).

271. Are 2.5 and 7.5 Hz the only two frequencies used? Authors should show a graph of the data spectra.

We modified this statement as we used every 0.25 Hz between 2.5 and 7.5 Hz (Lines 281-285 in the revised manuscript with track changes). We also added a table summarizing inversion parameters used at each inversion step (Supplementary Table 1).

275-276. Based on the V_p - Q values listed here, the authors have used $Q=90$ for $V_p=3.5-4.5$ km/s. However in cited reference 47, $Q=349$ for $V_p=3.6-4.6$ km/s. Please explain the discrepancy, and comment on any potential implications for the FWI results.

We used the Q value of 90 for accretionary prisms from reference 48 (Table 2 in Furumura et al., 2008) because we confirmed that this low Q value can reproduce synthetic waveforms that agree well with observed records. We also tested with a high Q value of 349 and found the inversion tend to be unstable very easily, so we derived only the Q value of 50 for the shallowest part of sediments from reference 47 (Hino et al., 2015). Hino et al. (2015) discuss that Q value of 349 from the Kumano Basin is much higher than other forearc areas, such as Costa Rica where Q_p is estimated to be 50 to 150 (Zhu et al., 2010). Hino et al. (2015) raise a

potential cause of the difference that the Kumano Basin is rigid (relatively stable) accretionary prism while the Costa Rica is an erosive margin associated with seamount subduction and thus host extensive fracturing in the forearc. We think the tectonic setting in the Hyuga-nada area is more similar as the case of Costa Rica.

279. Change “inverted.” to “updated.”. Authors should also show a figure illustrating the reduce in cost function.

We changed “inverted” to “updated” as suggested (Line 292 in the revised manuscript with track changes). For the reduction in cost function, it was difficult to illustrate it in a single figure or graph because we use different frequency values, offset range, and time windows at each inversion step. Instead of that, we describe it by the ratio of initial and final models at each inversion step together with other inversion parameters in Supplementary Table 1.

293. Please include the pre-stack depth migration velocity model in Extended Data figure 9.

We added the pre-stack depth migration velocity model to Supplementary Figure 9.

Figure 5a. In the caption, indicate that background color is V_p as in Fig. 1. The slab Moho dashed line shown in this figure has not been shown in previous figures nor mentioned in the text. Cite reference for the source of this Moho interface.

We added a statement to the caption of Figure 5a that indicate the background color is V_p in Figure 2a. We used iso- V_p contour of 7.5 km/s as a proxy of the slab Moho. We added this statement as well (Lines 578-579 in the revised manuscript with track changes).

This manuscript connects the occurrence of slow earthquakes with the presence of fluids in the upper plate. Some structures observed in the seismic image are interpreted as fluid migration paths based on the correlation with low-velocity zones observed in the velocity model and with the location of mud volcanoes. The zones interpreted as fluid-rich broadly correlate to the spatial distribution of tremors and very low-frequency earthquakes.

The manuscript is well-written and explanatory. However, the influence of the fluid

distribution in the seismogenic behaviour of subduction zones is already known and it is described in different publications (see a review in Saffer and Tobin, 2011; and Philip et al., 2023 for an example of a recent study in the Cascadia subduction zone). The changes in the reflectivity of the plate boundary reflector on seismic images were described in Ranero et al. (2008), and this reference should be included in the manuscript.

One of the novelties of this study is a discovery of vertical fluid conduits in the upper plate. This structure is obviously different from typical accretionary wedge system and, to our knowledge, has never been observed and discussed in any publications. In addition, we relate them to ridge subduction based on the comparison with seafloor bathymetry and sandbox experiments. We understand that many previous studies (including Ranero et al., 2008) have examined the relation between the seismic reflectivity and seismogenesis at the plate interface in subduction zones worldwide, but never been done in the Hyuga-nada area. Saffer and Tobin (2011) has been cited already (ref 3).

Thus, the main novelty of these results may be the direct comparison with the recorded seismicity in the area. However, the comparison of the seismic images with the earthquake location is only done spatially. For example, in Figure 5c, earthquakes within a large area are included, but the seismic profile shows high variability in the structure of the fluid migration system in distances <15 km. This variability in the structure should be integrated into the discussion of the seismogenic behaviour. To strengthen the results, I suggest focusing on the earthquakes occurring close to the seismic profile, and locating these earthquakes along the profile, to compare the crustal structure with the seismicity of the area. Also, nothing is mentioned about the depth of nucleation of these earthquakes. Are they nucleated at the plate boundary or at upper crust depths? If they are located at upper crust depths, why is there a difference driven by the fluid content of the plate boundary?

We have already incorporated most of the suggestions here in the original manuscript. To focus on the tremor and VLFE that occurred close to our seismic profile, we only used the events within 5-km distance from the seismic profile (See the caption of Figure 5b). Unfortunately, as the depth of these events are poorly constrained as explained by the cited references (Yamashita et al., 2015, 2021; Tonegawa et al., 2020), we cannot distinguish whether they occur at the plate boundary, in the upper plate or both. We do not suggest anywhere that the events

are located in the upper plate. All we can do is to discuss their spatial correlation in the horizontal directions as in the text.

Regarding the fluid's interpretation, there are fluids interpreted in the upper crust of the entire seismic profile, although there are only interpreted at the plate boundary towards the NE. The correlation of the low-velocity zones observed in the velocity model with mud volcanoes is unclear to me (Figure 3, see comments below). In my opinion, further discussion of the fluid origin and its implication for the seismogenic behaviour is needed.

In the southwestern part of the profile, the plate boundary is deeper than 15 km, which may be too deep for underthrust sediments to generate fluids via dehydration. To explain the slight difference in fluid distribution between the upper plate and the plate boundary, we added another possibility on the fluid origins, that is dehydration of clay-rich sediments within the upper plate, based on a new reference (Mitsutome et al. 2023, Scientific Reports) (Lines 190-192 in the revised manuscript with track changes).

Regarding the figures, I suggest revising Figure 3 to make it easier to understand. As it is, the relation of the low-velocity column with the mud volcanoes is difficult to see. Are there any mud volcanoes at the exact location of the low-velocity columns?

The relationship between the upper-plate low-velocity columns and the mud volcanoes on the seafloor is not straightforward because many of the low-velocity columns are still buried in the sedimentary layers (Fig. 2d-2i, 2e-2j, 2f-2k, 2g-2l). There is only one example forming a clear mound structure on the seafloor (Fig. 2h-2m). Figure 3 also aims to suggest that the study area hosts not only circular mounds indicating mud volcanoes but also a number of linear scarps associated with the ridge subduction, which is consistent with sandbox experiments (Dominguez et al. 1998).

In Figure 1 a, I can't find the green shaded area.

Two small green shaded areas are located just east of Kyushu Island.

Suggested bibliography:

Philip, B. T., Solomon, E. A., Kelley, D. S., Tréhu, A. M., Whorley, T. L., Roland, E., Tominaga, M., & Collier, R. W. (2023). Fluid sources and overpressures within the central Cascadia Subduction Zone revealed by a warm, high-flux seafloor seep. *Science*

Advances, 9(4). <https://doi.org/10.1126/sciadv.add6688>

Ranero, C. R., Grevemeyer, I., Sahling, H., Barckhausen, U., Hensen, C., Wallmann, K., Weinrebe, W., Vannucchi, P., Von Huene, R., & McIntosh, K. (2008). Hydrogeological system of erosional convergent margins and its influence on tectonics and interplate seismogenesis. *Geochemistry, Geophysics, Geosystems*, 9(3). <https://doi.org/10.1029/2007GC001679>

Saffer, D. M., & Tobin, H. J. (2011). Hydrogeology and mechanics of subduction zone forearcs: Fluid flow and pore pressure. *Annual Review of Earth and Planetary Sciences*, 39, 157–186. <https://doi.org/10.1146/annurev-earth-040610-133408>

REVIEWERS' COMMENTS

Reviewer #1 (Remarks to the Author):

Dear Editor,

the authors of manuscript NCOMMS-23-09256A have adequately addressed all the concerns and comments I had in my first review of this manuscript. I have no further comments on the revised manuscript. My recommendation is to publish the manuscript in its present form. Thank you for the opportunity to review this manuscript.

Reviewer #2 (Remarks to the Author):

I appreciate the effort that the authors have made to revise their manuscript. My comments have been tackled in the response letter. However, I still have some concerns mainly regarding the discussion of the interpretation and the seismogenic behavior. I understand that the length of the manuscript is limited, but from my point of view, some of the discussions are too short to fully convince the reader about the proposed interpretation. Below, there is a list of my detailed comments:

1) Given the high resolution of the data presented, I would expect a more detailed interpretation of the fluid migration paths and the correlation of their expression among the velocity model and the MCS image, as well as their relation to the bathymetric features. I do not necessarily disagree with the proposed interpretation in Figure 5a, but I think that further discussion is needed to fully justify the differences along the profile. For example, the sentence " The seismic data recorded in our study offer new evidence for the previously derived conclusions that the fluid source for the mud volcanoes in the Hyuga-nada subduction zone is along the plate boundary at depths >10 km" (lines 178-180 manuscript with track changes) is not in full agreement with the Figure 5a interpretation, as some of the fluid migration paths are interpreted as coming from the fluid-starved plate boundary (70-80 km) and also, the fluids interpreted between 15-35 km are associated with a fluid-starved plate boundary area and are interpreted as intra-crustal generated fluids. I would appreciate a longer discussion about the reasons behind this interpretation and the possibilities of the fluid origin. I think this is needed to justify your interpretation and it will strengthen your conclusions, as one of the main objectives of this manuscript is to improve the knowledge about fluid origin and migration paths.

2) I think that the high- V_p body in Figure 50 (0-30 km) is interpreted solely from the velocity model. Do you see any evidence of this body in the MCS? Is it somehow related to the fluid generation in this area?

3) For me, Figure 3 is not helping. As a suggestion, it would be nice to show the location of the mound that reaches the seafloor (figure 2h-m) on the map to show the correlation between both features. Also, I do not think that from figure 3 the straightforward conclusion is that the linear features are scarps due to the seamount subduction. I suggest revising this figure to clarify it.

4) Regarding the seismicity distribution and as a follow-up to my previous model about the earth-quake depth: A discussion on how the fluid distribution affects the seismicity is made. However, the distribution of fluid migration paths along the profile is very similar (see Figure 5a), and the main change is whether the plate boundary is fluid-rich or fluid-starved. However, the content of fluids at the plate boundary does not seem to be related to the presence or absence of fluid flows in the upper crust. Thus, I do not fully understand the discussion about the changes in the seismogenic behaviour in relation to the partial leakage of fluids into the upper plate. Also, the presence of the high- v_p velocity body (0-35 km) may have an effect on the seismicity distribution, and this possibility is not mentioned/discussed in the manuscript. In my opinion, a few more sentences about these issues will be helpful to further justify and explain your conclusions.

Point by point response to reviewer comments. Our responses are in **red bold**.

Reviewer #1 (Remarks to the Author):

Dear Editor,

the authors of manuscript NCOMMS-23-09256A have adequately addressed all the concerns and comments I had in my first review of this manuscript. I have no further comments on the revised manuscript. My recommendation is to publish the manuscript in its present form. Thank you for the opportunity to review this manuscript.

Reviewer #2 (Remarks to the Author):

I appreciate the effort that the authors have made to revise their manuscript. My comments have been tackled in the response letter. However, I still have some concerns mainly regarding the discussion of the interpretation and the seismogenic behavior. I understand that the length of the manuscript is limited, but from my point of view, some of the discussions are too short to fully convince the reader about the proposed interpretation. Below, there is a list of my detailed comments:

1) Given the high resolution of the data presented, I would expect a more detailed interpretation of the fluid migration paths and the correlation of their expression among the velocity model and the MCS image, as well as their relation to the bathymetric features. I do not necessarily disagree with the proposed interpretation in Figure 5a, but I think that further discussion is needed to fully justify the differences along the profile. For example, the sentence “ The seismic data recorded in our study offer new evidence for the previously derived conclusions that the fluid source for the mud volcanoes in the Hyuga-nada subduction zone is along the plate boundary at depths >10 km” (lines 178-180 manuscript with track changes) is not in full agreement with the Figure 5a interpretation, as some of the fluid migration paths are interpreted as coming from the fluid-starved plate boundary (70-80 km) and also, the fluids interpreted between 15-35 km are associated with a fluid-starved plate boundary area and are interpreted as intra-crustal generated fluids. I would appreciate a longer discussion about the reasons

behind this interpretation and the possibilities of the fluid origin. I think this is needed to justify your interpretation and it will strengthen your conclusions, as one of the main objectives of this manuscript is to improve the knowledge about fluid origin and migration paths.

Thank you for this suggestion. To respond to this comment, we added a sentence describing the relationship of the upper-plate conduits with the plate boundary (Lines 189-192 in the revised manuscript with track changes). For its relation to the bathymetry, please see our response to the third comment below and the updates of Figure 3.

2) I think that the high-Vp body in Figure 50 (0-30 km) is interpreted solely from the velocity model. Do you see any evidence of this body in the MCS? Is it somehow related to the fluid generation in this area?

Thank you for pointing this out. The high-Vp body shows relatively high reflectivity in the MCS image, but we are not sure if the change in reflectivity is related to fluid generation as it shows higher Vp. We only describe about the reflectivity in the text (Lines 157-165 in the revised manuscript with track changes).

3) For me, Figure 3 is not helping. As a suggestion, it would be nice to show the location of the mound that reaches the seafloor (figure 2h-m) on the map to show the correlation between both features. Also, I do not think that from figure 3 the straightforward conclusion is that the linear features are scarps due to the seamount subduction. I suggest revising this figure to clarify it.

We agree that the bathymetric features are so complicated that it is not easy to understand them only from the present form of Fig. 3. However, we believe that the linear features are primarily caused by the ridge subduction because their spatial pattern is similar to what is predicted by sandbox experiments that examine seamount subduction (ref 27). For this, we added some explanations to the text (Lines 135-136 in the revised manuscript with track changes). For better understanding of the spatial relationship between the subducting seamounts, upper-plate low-velocity zones and seafloor mounds and scarps, we also prepared a shaded relief map with iso-depth contours in Fig 3a. We added the locations of Fig. 2d-2h to the maps as suggested.

4) Regarding the seismicity distribution and as a follow-up to my previous model about the earth-quake depth: A discussion on how the fluid distribution affects the seismicity is made. However, the distribution of fluid migration paths along the profile is very similar (see Figure 5a), and the main change is whether the plate boundary is fluid-rich or fluid-starved. However, the content of fluids at the plate boundary does not seem to be related to the presence or absence of fluid flows in the upper crust. Thus, I do not fully understand the discussion about the changes in the seismogenic behaviour in relation to the partial leakage of fluids into the upper plate. Also, the presence of the high- v_p velocity body (0-35 km) may have an effect on the seismicity distribution, and this possibility is not mentioned/discussed in the manuscript. In my opinion, a few more sentences about these issues will be helpful to further justify and explain your conclusions.

Thank you also for this comment. We agree that the tremor distribution seems related only to the reflectivity (fluid contents) at the plate boundary and is not always relevant to the upper-plate low-velocity zones. To clarify this complexity and avoid potential confusions, we modified the statements (Lines 213-224 in the revised manuscript with track changes). Although the discussion on the correlation of high- V_p body in the western part (0-35 km distance) with the seismicity distribution has been made in the original manuscript, we modified the text for clarity (Lines 199-203 in the revised manuscript with track changes).